# Exogenous Nitric Oxide Promotes the Growth and Cadmium Accumulation of Alfalfa (*Medicago sativa*) Seedlings Under Cadmium Stress

**DOI:** 10.3390/plants14213264

**Published:** 2025-10-25

**Authors:** Yinping Chen, Yong Sun, Bo Cao, Maurice Ngabire, Yuzhi Lu, Qian Li, Qiaoling Yuan

**Affiliations:** School of Environmental and Municipal Engineering, Lanzhou Jiaotong University, Lanzhou 730070, China; s17339901136@163.com (Y.S.); l1670398136@163.com (B.C.); daweiwangabire@gmail.com (M.N.); lyz1720149102@163.com (Y.L.); 15719399651@163.com (Q.L.); yuanqq975@163.com (Q.Y.)

**Keywords:** nitric oxide, seedling growth, cadmium accumulation, antioxidant metabolism, *Medicago sativa*, heavy metal

## Abstract

As an important bioactive signaling molecule, nitric oxide (NO) participates in the responses of plants to various environmental stresses. The aim of this study was to investigate the influence of exogenous NO on the growth and cadmium (Cd) accumulation of alfalfa (*Medicago sativa*) during early growth. The results showed that Cd significantly inhibited alfalfa seedling growth and induced membrane lipid peroxidation. Addition of sodium nitroprusside (SNP, as an NO donor) significantly promoted seedling growth and induced the mobilization of seed photosynthate reserves, leading to an increase in total soluble sugar (SS) and reducing sugar (RS) contents. Application of SNP mitigated membrane peroxidation damage caused by Cd stress by enhancing catalase (CAT), ascorbate peroxidase (APX), superoxide dismutase (SOD) and peroxidase (POD) activities in order to eliminate reactive oxygen species (ROS), thereby improving Cd resistance and increasing Cd accumulation in alfalfa. This promoting effect of SNP depended on its concentration; the most optimal SNP concentration to promote the growth and Cd absorption of alfalfa under Cd stress was found to be 200 µM. The fresh weight (FW), dry weight (DW) and Cd accumulation of seedlings treated with 200 µM SNP increased significantly by 23.10%, 30.32% and 82.50%, respectively, on the fifth day, compared with the Cd-only treatment.

## 1. Introduction

Heavy metal contamination in soil has become increasingly serious because of rapid industrialization and more frequent anthropogenic activities and has created huge ecological risks [1]. The 2021 “China Ecological Environment Report” highlighted cadmium (Cd) as a main pollutant that decreases farmland soil quality [2].

The normal morphological and physiological development of plants can be impaired by Cd [3]. Depending on its concentration, Cd may either inhibit or stimulate the activity of antioxidant enzymes before any toxicity symptoms appear [4,5,6]. Reactive oxygen species (ROS), such as superoxide radicals (O_2_·^−^) and hydrogen peroxide (H_2_O_2_), are produced by metabolically active cells in seedlings, especially during early growth [7]. Cells are equipped with enzymatic mechanisms to eliminate ROS or mitigate the oxidative damage they cause [8]. Superoxide dismutase (SOD) and peroxidase (POD) convert O_2_·^−^ into H_2_O_2_; then, H_2_O_2_ is decomposed to H_2_O and O_2_ by catalase (CAT) and ascorbate peroxidase (APX) [3]. Seedlings exposed to Cd stress exhibit disorders in carbohydrate mobilization, damaged membrane, impaired translocation of food reserves, increased relative electric conductivity (REC) and decreased root and shoot growth [9,10]. However, ROS also have beneficial functions. For example, in the root system of rice (*Oryza sativa*), ROS signals work with Ca signals synergistically to increase more root biomass in non-stress areas, thereby avoiding local Cd stress [11].

Nitric oxide (NO), as a small, gaseous, bioactive signaling molecule, can diffuse into cells readily, where it interacts with various cellular compounds [12]. NO plays a key role in various processes of plant growth and development, as well as in plant responses to many kinds of stresses, including Cd stress [13,14]. For example, sodium nitroprusside (SNP, an NO donor) alleviates the toxic effects of As on the growth of mung bean (*Vigna radiata*) seedlings subjected to arsenic (As) stress by enhancing plasma membrane integrity, decreasing As accumulation, and increasing hydrolytic enzyme activities [15]. In addition, NO also alleviates Cd toxicity by promoting the uptake of nutrients and reducing Cd translocation from root to shoot by down-regulating *OsLsi2* transporter expression. *OsLsi2* restricts Cd transport from root to shoot via many pathways such as substrate competition, cellular compartmentalization and signal regulation [16,17]. In contrast, NO can intensify Cd toxicity by up-regulating the expression of related genes to promote Cd uptake and accumulation [12]. For example, Cd accumulation in thale cress (*Arabidopsis thaliana*) roots was enhanced by NO through up-regulating the expression of Fe uptake-related genes *IRT1*, *FRO2* and *FIT* [18]. These contradictory conclusions show that the role of NO is highly controversial [19]. Collectively, NO may have a dual effect in plant responses to Cd; furthermore, the role of NO depends on NO donor concentration. Low concentrations of SNP can mitigate Cd stress more effectively than high concentrations [20]. As is known, heavy metal pollution remediation plants need to have both resistance and accumulation capacity for heavy metals. It is worth noting that NO induced not only Cd tolerance but also phytoremediation potential of mustard (*Brassica juncea*) [21], thale cress [22], Chinese cabbage (*Brassica rapa*) [23] and millet (*Setaria italica*) [24]. Therefore, it is important to investigate the role of NO in Cd pollution phytoremediation.

Phytoremediation has been well advocated over the past decade. Researchers have begun studying the use of plants such as alfalfa (*Medicago sativa*) to remove heavy metals such as Cd from contaminated soil [25]. Alfalfa is extremely resistant to heavy metals and functions as a strong bio-accumulator [26]. Although the role of the NO donor in alleviating heavy metal toxicity has been elucidated in recent studies [12], its influence on seedling growth, ROS metabolism and Cd accumulation of alfalfa under Cd stress has not yet been examined. It is necessary to determine whether NO can induce the Cd tolerance of alfalfa and the optimal concentration of the NO donor for improving the remediation efficiency of alfalfa. This is of great significance in promoting the practical application of NO donors in phytoremediation.

In a previous study, NO exhibited dual functions in plant responses to Cd, and the different role of NO depended on the NO donor concentration. However, the function of NO in alfalfa responses to Cd and the specific action mechanism are still not clear. Therefore, the hypothesis of this study is that an appropriate concentration of SNP will exert protective functions in alfalfa exposed to Cd. We aimed to determine whether NO could promote alfalfa growth and Cd uptake and clarify the physiological and biochemical mechanisms of NO action. For this purpose, the changes in biomass, ROS metabolism and Cd accumulation in seedlings treated with 25 to 500 µM SNP under 30 µM Cd stress were analyzed. We found that SNP promoted alfalfa growth and Cd uptake. Furthermore, this promoting effect is linked to the major antioxidant enzyme activities and storage substance contents. However, this positive effect depended on the concentration of SNP. Therefore, we also aimed to identify an optimal SNP concentration for alleviating Cd toxicity and improving the remediation efficiency of alfalfa to Cd pollution. This study is the first to investigate the effect of NO on alfalfa growth and Cd accumulation during early growth and to clarify the physiological and biochemical mechanisms of NO action. It will provide a theoretical basis and new ideas for alfalfa application in Cd pollution remediation.

## 2. Results

### 2.1. Seedling Growth

To evaluate how SNP affects seedling growth in Cd stress, seedling fresh weight (FW) and dry weight (DW) were measured. To illustrate the differences among various treatments more clearly, Figure 1 shows the growth state of a single seedling in each treatment after 5 days. The growth conditions of all seedlings are shown in Appendix A.

In the present study, compared with the control (CK), FW and DW decreased significantly by 24.58% and 18.00%, respectively, on the fifth day, under 30 µM Cd stress (Figure 2A,B). Application of SNP overcame the negative impact of Cd stress. Alfalfa growth increased significantly under Cd stress. All SNP treatments significantly improved alfalfa growth compared with the Cd-only treatment. In addition, 200 µM SNP conferred the greatest protection against Cd toxicity; FW and DW increased significantly by 23.10% and 30.32%, respectively, compared with the Cd-only treatment (Figure 2).

As the main energy substances for germination and growth of plant seeds, the contents of soluble sugars (SS) and reducing sugars (RS) significantly increased, especially by day 2 or 3 after treatment, and these higher contents were maintained in the following days (Figure 3A,B). Under Cd stress, SS and RS contents were lower than those in CK throughout the whole experiment. In contrast, lower levels of SNP (25 to 200 µM) significantly increased SS and RS contents in alfalfa seeds under Cd stress, especially by day 2 or 3 after treatment with 200 µM SNP. However, higher levels of SNP (300 to 500 µM) did not exhibit this effect, especially at an SNP level of 500 µM (Figure 3). These findings clearly demonstrate that lower levels of SNP enhanced the breakdown of storage substances, resulting in increased RS and SS contents to provide energy for seedling growth under Cd stress.

### 2.2. Antioxidant Metabolism

It is well known that oxidative damage triggered by Cd stress sparks an excessive accumulation of ROS, which causes lipid peroxidation. In this study, compared with CK, 30 µM CdCl_2_ stress increased malondialdehyde (MDA) content on the second, third and fifth day significantly. Similarly, for REC, production of H_2_O_2_ and O_2_ increased on the first, third and fourth day significantly (Figure 4). Under Cd stress, SNP (>25 µM) application increased MDA content significantly on day 1 (Figure 4A). However, SNP decreased MDA content significantly compared with the Cd-only treatment from the second day to the fifth day. A similar behavior was found for REC (Figure 4B). Under Cd stress, SNP (200 or 300 µM) decreased H_2_O_2_ and O_2_·^−^ levels in the first 3 days. However, SNP (>300 µM) increased H_2_O_2_ and O_2_·^−^ levels to values similar to or above those in the Cd-only treatment on the fifth day (Figure 4C,D).

In the first two days, SNP (25 to 200 µM) increased SOD and CAT activities significantly, reaching the highest value when SNP concentration was 200 µM. However, higher SNP concentrations (>200 µM) decreased SOD and CAT activities significantly (Figure 5A,C). The POD activity increased for the first 3 days, then stabilized and reached the highest value when SNP concentration was 200 µM on the third, fourth and fifth day (Figure 5B). The activity of APX increased on the first day, especially for SNP (25 to 100 µM), although its activity decreased steadily thereafter with an increase in SNP concentration (Figure 5D). These results show that SNP can maintain the high activity of different antioxidant enzymes in the early stages of plant growth to eliminate ROS and alleviate membrane peroxidation damage caused by Cd stress.

### 2.3. Cd Accumulation

Compared with CK, Cd content and accumulation in seedlings increased significantly after 2 days of 30 µM Cd stress (Figure 6A,B). Application of SNP increased the content and accumulation of Cd in a concentration-dependent manner under Cd stress. When the SNP concentration was lower than 200 µM, Cd content and accumulation increased significantly with an increase in SNP concentration. However, when the SNP concentration was higher than 200 µM, the effect of SNP in promoting Cd absorption and accumulation gradually weakened with an increase in SNP concentration. The maximum Cd content and accumulation was reached when SNP concentration was 200 µM (Figure 6).

### 2.4. Effect of 200 µM SNP on the Growth and Physiological Metabolism of Alfalfa

Based on the aforementioned results, subsequent experiments were conducted in which seedlings were treated with only the optimal concentration of SNP (200 µM) to evaluate its individual effects on seedling growth in the absence of Cd (Table 1). The optimal concentration of SNP promoted seed germination and seedling growth. Compared with CK, germination percentage, germination index, seedling vigor index and germination energy increased by 36.06%, 66.18%, 118.13% and 66.95%, while root length, shoot length, FW and DW increased by 27.00%, 36.22%, 32.14% and 44.00%, respectively. At the same time, 200 µM SNP enhanced the breakdown of storage substances, compared with CK, and RS increased by 25.12%. Furthermore, compared with CK, 200 µM SNP protected seedling cells against oxidative damage by decreasing MDA, H_2_O_2_ and O_2_·^−^ contents by 56.55%, 68.48% and 50.00%, respectively. Compared with CK, the activities of POD, SOD, CAT and APX in seedlings were reduced after being treated with 200 µM SNP. It is possible that there are other antioxidant enzymes and antioxidants to eliminate ROS.

### 2.5. Principal Component and Pearson Correlation Analyses

Principal component analysis (PCA) and correlation analysis are commonly used statistical methods in the study of influencing factors. As mentioned earlier, under Cd stress, SNP has a positive effect in promoting seedling growth and Cd accumulation of alfalfa. To further investigate the mechanism of SNP action, we conducted PCA and Pearson correlation analysis on seedling growth, various physicochemical indicators and Cd accumulation in alfalfa under 30 µM Cd stress on the fifth day (Figure 7A). The maximum contribution was exhibited by Dim1 and Dim2, explaining more than 67.4% of the variance, with Dim1 at 54.3% and Dim2 at 13.1%. All parameters indicated that alfalfa growth and physiology were significantly affected by Cd stress. The first principal component captured 54.3% of the total variance in physicochemical parameters, with DW, POD, Cd content (Cd-c), Cd accumulation (Cd-a), SS, FW and RS being positively correlated with the first principal component, while SOD, CAT, APX, hydrogen peroxide (HP, H_2_O_2_) and superoxide radicals (SR, O_2_·^−^) being negatively correlated (Figure 7A).

There was an extremely significant positive correlation of Cd content and accumulation in alfalfa seedlings with DW, FW, SS, RS and POD (*p* = 0.0001 and *p* = 0.0001), but a significantly negative correlation with SR (*p* = 0.0376 and *p* = 0.0341) and an extremely significant negative correlation with HP, SOD, CAT and APX (*p* = 0.0001 and *p* = 0.0001). There was a significant positive correlation of seedling growth parameter DW with POD and SS (*p* = 0.0131 and *p* = 0.0195), but a significant negative correlation with HP and MDA (*p* = 0.0194 and *p* = 0.0102) and an extremely significant negative correlation with APX, SOD and SR (*p* = 0.0001, *p* = 0.0001, *p* = 0.0010, respectively). Similarly, there was a significant positive correlation of FW with SS and RS (*p* = 0.0014 and *p* = 0.0112) and an extremely significant positive correlation with POD (*p* = 0.0001), but a significant negative correlation with SR (*p* = 0.0029) and an extremely significant negative correlation with APX, CAT, SOD and HP (*p* = 0.0001, *p* = 0.0001, *p* = 0.0001, *p* = 0.0003, respectively) (Figure 7B).

## 3. Discussion

Seedling growth is inhibited significantly by Cd stress [3]. This can be attributed to deaccelerated breakdown of storage substances, resulting in decreased RS and SS contents, which are essential as fuel that provides energy for seedling growth [10]. Application of SNP alleviates the inhibition of seedling growth attributed to Cd stress in a dose-dependent manner [27]. Amylase activities can be strongly stimulated by SNP to increase the contents of RS and SS in tomato (*Solanum lycopersicum*) [28] and sesame (*Sesamum indicum*) [29] seedlings. However, a higher concentration of SNP decreases them. This may be related to the generation of peroxynitrite (ONOO^−^), which damages cell membranes and restrains some potential antioxidant enzymes, and as result, ROS accumulates [30,31]. From the present results, as in many other plant species, SNP significantly improved growth of alfalfa under Cd stress (Figure 1 and Figure 2). Similarly, a lower concentration of SNP (25 to 200 µM) significantly increased SS and RS content in alfalfa seedlings under Cd stress, especially by day 2 or 3 after treatment (Figure 3). Our results indicate that SNP can strongly stimulate amylases to break down starch stored in endosperm into glucose, providing direct energy for cell division, respiration and seedling growth. Furthermore, the accumulated reducing sugars can lower the osmotic potential of cells, maintain normal cell metabolism and thus adapt to Cd stress. Addition of 200 µM SNP produced the greatest protection of alfalfa against Cd toxicity. However, higher SNP levels of 300 to 500 µM did not exhibit this positive effect.

Excessive Cd can induce lipid peroxidation and membrane damage [32]. In this study, 30 µM CdCl_2_ treatment increased REC and MDA contents and resulted in higher production of H_2_O_2_ and O_2_·^−^ in the alfalfa seedlings (Figure 4). This suggested that Cd has caused significant lipid peroxidation during early growth. This lipid peroxidation results in reduced fluidity of membranes, increased membrane leakiness, damaged membrane proteins and inactivation of receptors, enzymes and ion channels [33,34,35]. NO can restore and protect cell membranes from lipid peroxidation damage. In our study, the increased activities of anti-oxidative enzymes after application of SNP (Figure 5) confirmed previous reports in rice [36] and perennial ryegrass (*Lolium perenne*) seedlings [37]. These results indicate that SNP has a positive effect in protecting seedling cells against oxidative damage by increasing antioxidant enzyme activities, scavenging ROS and decreasing MDA accumulation [21]. Interestingly, in the cells, NO at lower levels is a positive mediator when plants respond to stresses, including metals stress [38]. On the contrary, NO at high levels accentuates cell damage because of nitrosative stress [39]. At specific concentrations, NO activates antioxidant systems to eliminate ROS, thus decreasing oxidative damage [40]. In this study, 200 µM SNP regulates the alfalfa antioxidant system to protect membrane system stability and reduce Cd damage during early growth.

In our study, alfalfa had a better ability of accumulating Cd when treated with SNP (Figure 6), which was in accordance with the results of Wang et al. [26]. The positive effect of NO in Cd accumulation has been reported in *Arabidopsis* [22], Chinese cabbage [23] and millet [24]. NO promotes Cd uptake and accumulation by up-regulating *IRT1*, *FRO2* and *FIT* gene (Fe uptake-related) expression under Cd stress [18]. However, the positive role, in relation to phytoremediation, of NO depends on the concentration of SNP. Similarly, in our study, the NO promoting effect on Cd accumulation reached a maximum when the SNP concentration was 200 µM. Subsequently, with an increase in SNP concentration, the promoting effect weakened (Figure 6). This may be dependent on the fact that NO can react with ROS to produce more toxic oxides such as ONOO^−^ [41,42]. Depending on its concentrations in cells, NO exerts both beneficial and harmful effects on plants [43]. Thus, it is essential for living organisms to control NO levels strictly to tune a fine balance between beneficial and harmful actions of NO. However, NO significantly reduced Cd and As uptake in wheat (*Triticum aestivum*) and rice [16,44]. One possible reason is that Cd and As accumulation decreases in roots through the down-regulation of *OsLsi1* transporter expression, which in turn decreases Cd and As accumulation in shoots through the down-regulation of *OsLsi2* transporter expression, thereby affecting the xylem loading of Cd and As through a co-transport mechanism and indirectly regulating Cd transport from roots to shoots [45]. Another possible reason is that SNP improves plant resistance to Cd by increasing membrane transporter activity, which prevents Cd/As from entering root cells, but is conducive to the absorption of some mineral nutrients, such as Ca^2+^ and K^+^. In our study, 200 µM SNP did not exhibit negative effects on alfalfa but instead promoted its growth in the absence of Cd by enhancing the breakdown of storage substances and decreasing MDA, H_2_O_2_ and O_2_·^−^ contents (Table 1). Furthermore, under Cd stress, 200 µM SNP regulated the alfalfa antioxidant system to protect membrane system stability, increased alfalfa’s resistance to Cd, promoted seedling growth and thus accumulated more Cd. These different results indicate that NO may have a dual action on heavy metal accumulation in plants by regulating expression of genes related to heavy metal absorption and transport.

Principal component analysis and Pearson correlation analysis revealed that membrane peroxidation occurred in alfalfa under Cd stress (Figure 7). Analysis of MDA, SR and HP, as indicators of cell plasma membrane damage, also provided strong evidence for the damage of alfalfa cells under Cd stress. Accordingly, the growth parameters (DW, FW), Cd content and its accumulation in alfalfa seedlings were negatively correlated with MDA, HP and SR. This indicated that cell plasma membrane damage inhibited the growth and Cd accumulation of alfalfa. In contrast, the growth parameters, Cd content and its accumulation were positively correlated with SS, RS and POD. The induction of POD activity and the synthesis and accumulation of SS and RS were important resistance mechanisms of alfalfa in response to Cd stress. The presence of exogenous NO at specific concentrations activated enzymatic and non-enzymatic antioxidant systems such as POD, SS and RS to improve tolerance against Cd toxicity, promoted seedling growth and increased the accumulation of Cd in alfalfa; however, excessive levels of NO accentuated cell damage by inducing nitrosative stress (Figure 8). Thus, like a double-edged sword, NO exerts a dual function on alfalfa growth and Cd accumulation under Cd stress, and this dual function of NO might depend on its concentration in cells [42,43]. Additionally, the dual function of NO stems from its multi-target characteristics. NO can not only activate defense genes (such as antioxidant defense-related genes and osmotic regulation-related genes) but also amplify stress damage through a free radical chain reaction. This contradiction indicates that when analyzing the role of NO, it may be necessary to take into account the plant species and synergistic effect of NO with other signals (ROS, Ca^2+^ and phytohormone). The imbalance in the communication between signaling molecules may be the fundamental cause of the dual effects.

Besides SNP, S-nitroso-N-acetylpenicillamine (SNAP) is another donor of exogenous NO [43]. The endogenous NO produced in plants is mainly catalyzed by enzymes with NO synthase (NOS) activity, nitrate reductase (NR) and nonenzymatic reactions [14,42]. In order to confirm and obtain a more profound insight about the regulatory effect of NO on alfalfa growth and Cd accumulation, it is necessary to conduct experiments using SNP, SNAP, NO scavengers (2-(4-carboxyphenyl)-4,4,5,5-tetramethylimidazoline-l-oxyl-3-oxide, cPTIO), NOS inhibitor (L-nitro-arginine methylester, L-NAME) and NR inhibitor (NaN_3_). In subsequent research, molecular biology and metabolomics methods should also be used to acquire a comprehensive and thorough understanding of the underlying mechanism of NO-induced alfalfa tolerance to Cd.

## 4. Materials and Methods

### 4.1. Experimental Design and Plant Growth Conditions

A pre-experiment revealed that the concentration of Cd (administered as CdCl_2_·2.5 H_2_O) required to reduce germination percentage by 50% was 30 µM, so we chose this as the exposure level in the present study. Based on preliminary experiments, 0–500 µM was selected as the SNP treatment concentration, which can not only effectively regulate NO signal but also avoid toxicity risk. Alfalfa seeds were provided by Gansu Academy of Agricultural Sciences, China. Sodium hypochlorite (NaOCl, 2%) was used to surface-sterilize seeds for 5 min, which were then thoroughly rinsed with distilled water. The seeds were cultivated on single-layer filter paper (diameter: 10 cm) in Petri dishes (diameter: 10 cm), and 10 mL one-half Hoagland’s nutrient solution was used to moisten the filter paper. Cd as CdCl_2_·2.5 H_2_O and NO as sodium nitroprusside (SNP, in the form of C_5_FeN_6_Na_2_O·2H_2_O,) were added to the nutrient solution [30]. The treatment combinations were as follows: (1) control (CK); (2) 30 µM Cd (Cd); (3) 25 µM SNP + 30 µM Cd (25SNP + Cd); (4) 50 µM SNP + 30 µM Cd (50SNP + Cd); (5) 100 µM SNP + 30 µM Cd (100SNP + Cd); (6) 200 µM SNP + 30 µM Cd (20SNP + Cd); (7) 300 µM SNP + 30 µM Cd (300SNP + Cd); (8) 400 µM SNP + 30 µM Cd (400SNP + Cd); (9) 500 µM SNP + 30 µM Cd (500SNP + Cd). We used three replicates of each treatment every day. Each replicate had about at least 600 seeds, and there were at least 30 seeds in each Petri dish. The dishes were incubated at 25 ± 1 °C with 80 percent relative humidity, 16 h of light (6500 lx, T8LED, Chongqing Qingmu Tissue Culture Technology Co., Ltd., Chongqing, China) and 8 h dark every day in the culturing box (LRH-250-G, Guangdong Taihongjun Scientific Instruments Co., Ltd.; Shaoguan City, Guangdong Province, China) for 5 days. The solutions in Petri dishes were renewed every day in order to keep a stable concentration. Seedling samples were taken to measure various indicators every day.

Based on the results of the above experiments, an optimal SNP concentration for alleviating Cd toxicity and improving alfalfa Cd accumulation was identified. Subsequent experiments were conducted in which alfalfa seedlings were treated with this optimal concentration of SNP to evaluate the individual effects of SNP on alfalfa growth in the absence of Cd. Seedlings were treated and cultured as described in the above experiments for 5 days. Afterwards, the seedlings were taken to measure various parameters.

### 4.2. Growth Parameter Determination

Germination percentage, germination index, seedling vigor index, germination energy, root length and shoot length were assayed according to the method developed in [46]. Before measuring FW, we washed the seedlings with deionized water and then dried the surface moisture with filter paper. To determine DW, the seedlings were dried at 105 °C for 30 min and then dried at 80 °C until a constant weight was achieved. FW and DW per 100 seedlings were measured every day for 5 days.

### 4.3. Lipid Peroxidation Determination

Lipid peroxidation was analyzed by determining REC and the content of H_2_O_2_, MDA and O_2_·^−^.

REC and H_2_O_2_ were determined according to the method of [47]. Seedlings were sectioned into small pieces, thoroughly rinsed and subsequently immersed in deionized water for 30 min. After measuring the initial conductivity (Ci), the samples were boiled for 15 min, and the final conductivity (Cf) was measured at 25 °C. REC was calculated according to the following formula: REC = (Ci/Cf) × 100%. Seedlings (0.3 g) were ground in liquid nitrogen together with 3 mL 0.1% trichloroacetic acid (TCA). Then, they were centrifuged at 12,000× *g* for 15 min. A mixture consisting of the supernatant, 1 M KI and 10 mM potassium sulfate buffer was vortexed. The absorbance at 390 nm was measured with 0.1% TCA as a blank. The content was calculated by H_2_O_2_ standard curve.

MDA was determined as described by [48]. Fresh seedlings (0.3 g) were ground in 3.0 mL 5% TCA, and then centrifuged at 12,000× *g* for 15 min. A mixture consisting of the supernatant, 20% TCA and 0.5% thiobarbituric acid was heated at 95 °C for 30 min, and then centrifuged at 7500× *g* for 5 min after cooling. The absorbance at 532 nm was measured and corrected by subtracting the absorbance at 600 nm.

The O_2_·^−^ content was determined using the hydroxylamine oxidation method [49] with some modifications. Fresh seedlings (0.5 g) were ground with 3.0 mL of 65 mM potassium phosphate buffer (pH 7.8), and then centrifuged at 10,000× *g* for 10 min. A mixture consisting of the supernatant and 0.5 mM hydroxylamine chloride was warmed for 20 min at 25 °C. P-aminobenzene sulfonic (5 mM) and α-naphtylamine (7 mM) were added. The mixture was warmed again for 20 min at 25 °C and 4.0 mL of Diethyl-ether was added. The absorbance at 530 nm was measured. A standard curve generated using nitrogen dioxide radical (NO_2_^−^) was used to calculate O_2_·^−^ content.

### 4.4. Antioxidant Enzyme Extraction and Assays

Fresh seedling tissues (0.5 g) were used to extract enzymes using an extraction buffer containing 50 mM PO_4_^3−^ buffer (pH 7.8), 0.2 mM ethylene diaminete traacetic acid (EDTA) and 2.0% polyvinyl-pyrrolidone [50].

One SOD unit was the quantity of enzyme used to restrain the photoreduction of nitro-blue tetrazolium (NBT) by 50% at 25 °C [50]. The reaction system (NBT, L-methionine, EDTA, riboflavin, sodium carbonate and enzyme extract) was maintained in light and dark for 15 min each; then, absorbance at 560 nm was measured.

H_2_O_2_ was added to the reaction solution (PO_4_^3−^ buffer, enzyme extract) to start the reaction. CAT activity was represented by a decrease in absorbance at 240 nm within 30 s [51].

The POD activity was recorded by an increase in absorbance at 470 nm attributed to guaiacol oxidation [52]. Enzyme extract, PO_4_^3−^ buffer, guaiacol and H_2_O_2_ were mixed as the assay solution. The change in absorbance was observed at 1-minute intervals for 3 mins. A change of 0.01 in absorbance per minute was taken as one unit of POD activity.

The APX activity was recorded based on decrease in absorbance at 290 nm caused by ascorbic acid oxidation [53]. The enzyme extract, PO_4_^3−^ buffer, EDTA, ascorbic acid and H_2_O_2_ were mixed as the assay solution. The change in absorbance was observed for 60 s. The quantity of enzyme used to oxidize 1 µM of ascorbic acid min^−1^ at 25 °C was taken as one unit of APX activity.

U·g^−1^ FW·min^−1^ was used to express enzyme activity.

### 4.5. Storage Substances Determination

The total SS was estimated using anthrone reagent [54]. RS was determined using DNS (3,5-Dinitrosalicylic acid) reagent [55]. The amounts of SS and RS were calculated using a standard curve prepared from glucose.

### 4.6. Cd Concentration Determination

For Cd analysis, 100 seedlings were selected at random from each replicate every day. Cd concentration was determined using HNO_3_ and HClO_4_ as described by [56]. Each group (comprising twenty samples) included two standard reference samples (HTSB-2 and NST-2) and one blank control sample. The recovery rates of the internal standard responses across all samples exceeded 90%. The dry plant tissues were ground and then passed through an 80-mesh sieve. We took a 0.2 g ground sample and placed it in a digestion tube, added 5 mL HNO_3_, and then let it stand for 8 h. The digestion tube was heated in a digestion instrument at 120 °C for 3 min. After cooling to room temperature, 2 mL HClO_4_ was added. The temperature was raised to 180 °C again and heating continued for digestion until the solution was clear and transparent with a light yellow color. The sample was diluted to 25 mL with ultrapure water, and the content of Cd in plant tissues was measured using a Flame Atomic Absorption Spectrophotometer (220 Thermo Nicolet Corporation, Waltham, MA, USA). The total Cd accumulation was calculated on the basis of Cd concentration and plant biomass.

### 4.7. Statistical Analysis

Data were expressed in the form of means ± standard deviations (SDs). There were three independent experiments, and each experiment was replicated three times (*n* = 9). One-way analysis of variance (ANOVA) and Tukey’s test (HSD) were performed using statistical package STATISTICA 7 to analyze the data. Different letters show significant differences between treatments at *p* < 0.05. Logarithmic or inverse transformations were used to standardize the data. The relationships among different variables were quantified using PCA. Correlation coefficient calculation and principal component analysis were performed using RStudio version 2024.12.0+467.

## 5. Conclusions

The present research shows that 30 µM Cd noticeably reduced the growth of alfalfa seedlings, likely due to the ability of Cd to inhibit the mobilization of carbohydrates and other nutrients and intensify oxidative stress. Application of SNP significantly mitigated Cd toxicity in a concentration-dependent manner by activating antioxidant enzymes to eliminate ROS. SNP application also improved seedling growth under Cd stress by enhancing the breakdown of storage substances. The optimal concentration of SNP to significantly mitigate Cd toxicity, promote seedling growth and then increase the accumulation of Cd in alfalfa appears to be 200 µM.

## Figures and Tables

**Figure 1 plants-14-03264-f001:**
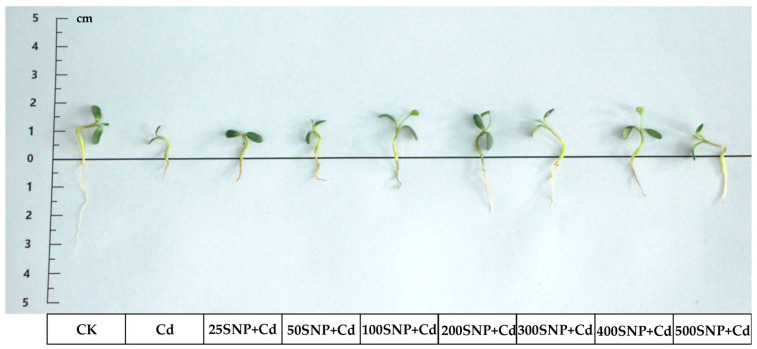
Alfalfa seedlings treated with different concentrations of SNP after 5 days under 30 µM Cd stress (seedlings were cultured in 1/2 Hoagland solution in an LRH-250-G climatic chamber (Guangdong Taihongjun Scientific Instruments Co., Ltd.; Shaoguan City, Guangdong Province, China), which was maintained at a constant temperature of 25 ± 1 °C, a relative humidity of 80 ± 5%, and a photoperiod of 16 h light (6500 lx, T8LED, Chongqing Qingmu Tissue Culture Technology Co., Ltd., Chongqing, China) and 8 h dark every day.

**Figure 2 plants-14-03264-f002:**
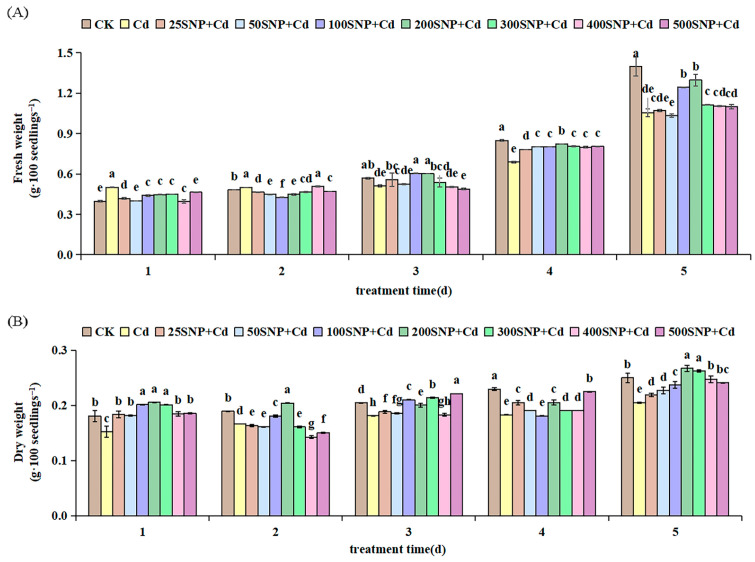
Effects of SNP on (**A**) fresh weight and (**B**) dry weight of alfalfa seedlings under 30 µM Cd stress. Data are expressed as mean ± SD (standard deviation) (*n* = 9). Values were compared using ANOVA, and when results were significant, the treatments were compared using Duncan’s test (*p* < 0.05). For a given day, bars labeled with different letters differ significantly.

**Figure 3 plants-14-03264-f003:**
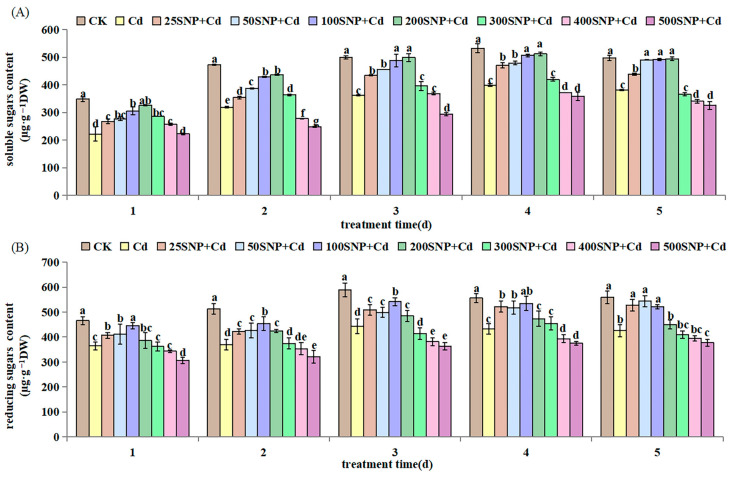
Effects of SNP on the contents of (**A**) total soluble sugars and (**B**) reducing sugars in alfalfa seedlings under 30 µM Cd stress. Data are expressed as mean ± SD (standard deviation) (*n* = 9). Values were compared using ANOVA, and when results were significant, the treatments were compared using Duncan’s test (*p* < 0.05). For a given day, bars labeled with different letters differ significantly.

**Figure 4 plants-14-03264-f004:**
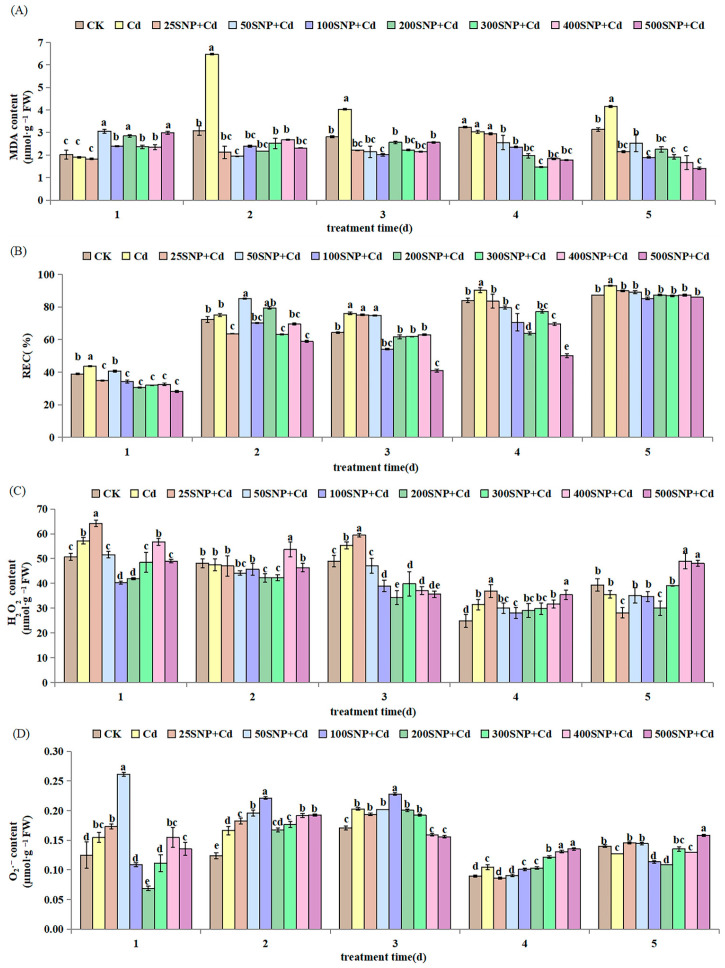
Effect of SNP on the content of (**A**) MDA, (**B**) REC, (**C**) H_2_O_2_ and (**D**) O_2_·^−^ in alfalfa seedlings under 30 µM Cd stress. Data are expressed as mean ± SD (standard deviation) (*n* = 9). Values were compared using ANOVA, and when results were significant, the treatments were compared using Duncan’s test (*p* < 0.05). For a given day, bars labeled with different letters differ significantly.

**Figure 5 plants-14-03264-f005:**
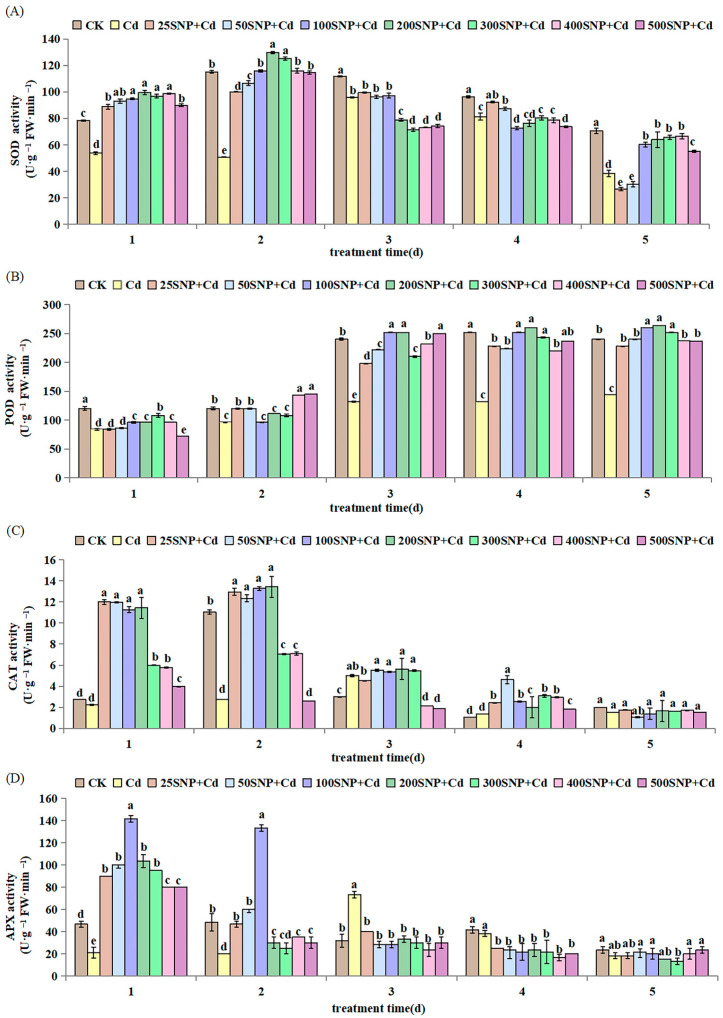
Effect of SNP on the activity of (**A**) SOD, (**B**) POD, (**C**) CAT and (**D**) APX in alfalfa seedlings under 30 µM Cd stress. Data are expressed as mean ± SD (standard deviation) (*n* = 9). Values were compared using ANOVA, and when results were significant, the treatments were compared using Duncan’s test (*p* < 0.05). For a given day, bars labeled with different letters differ significantly.

**Figure 6 plants-14-03264-f006:**
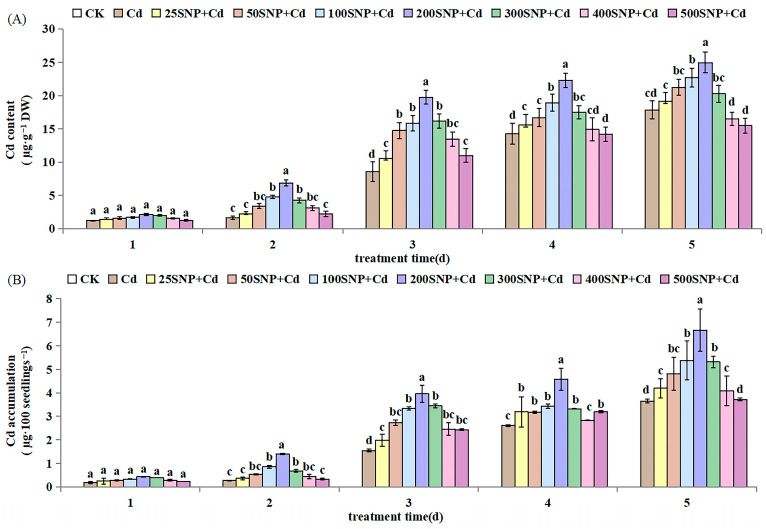
Effect of SNP on the (**A**) content and (**B**) accumulation of Cd in alfalfa seedlings under 30 µM Cd stress. Data are expressed as mean ± SD (standard deviation) (*n* = 9). Values were compared using ANOVA, and when results were significant, the treatments were compared using Duncan’s test (*p* < 0.05). For a given day, bars labeled with different letters differ significantly.

**Figure 7 plants-14-03264-f007:**
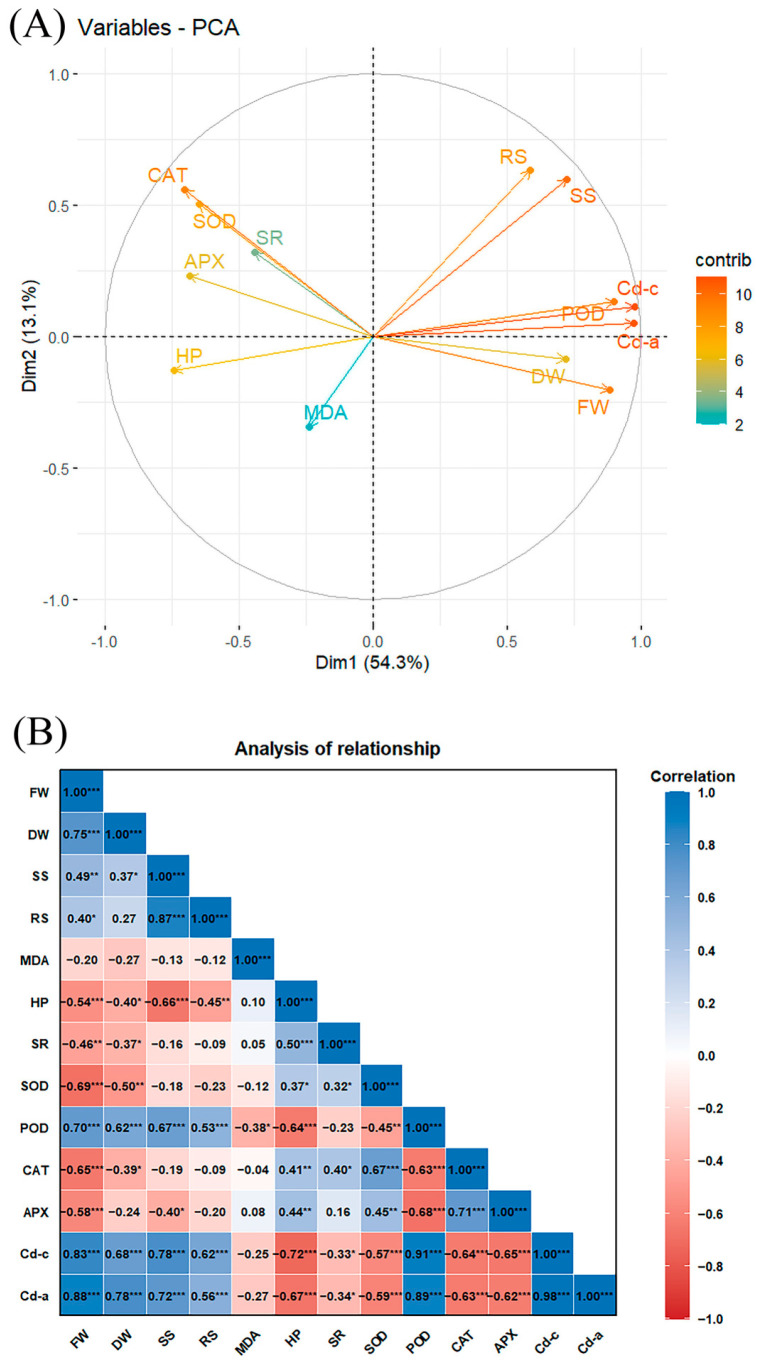
(**A**) Loading plot of principal component analysis (PCA) and (**B**) correlation of various attributes in alfalfa treated with different levels of SNP under 30 µM Cd stress. Different abbreviations used in the figure are as follows: FW (fresh weight), DW (dry weight), SS (soluble sugars), RS (reducing sugars), MDA (malondialdehyde), HP (hydrogen peroxide), SR (superoxide radicals), SOD (superoxide dismutase), POD (peroxidase), CAT (catalase), APX (ascorbate peroxidase), Cd-c (Cd content), and Cd-a (Cd accumulation). * Significant correlation at 0.05 level (both sides); ** significant correlation at 0.01 level (both sides); *** significant correlation at 0.001 level.

**Figure 8 plants-14-03264-f008:**
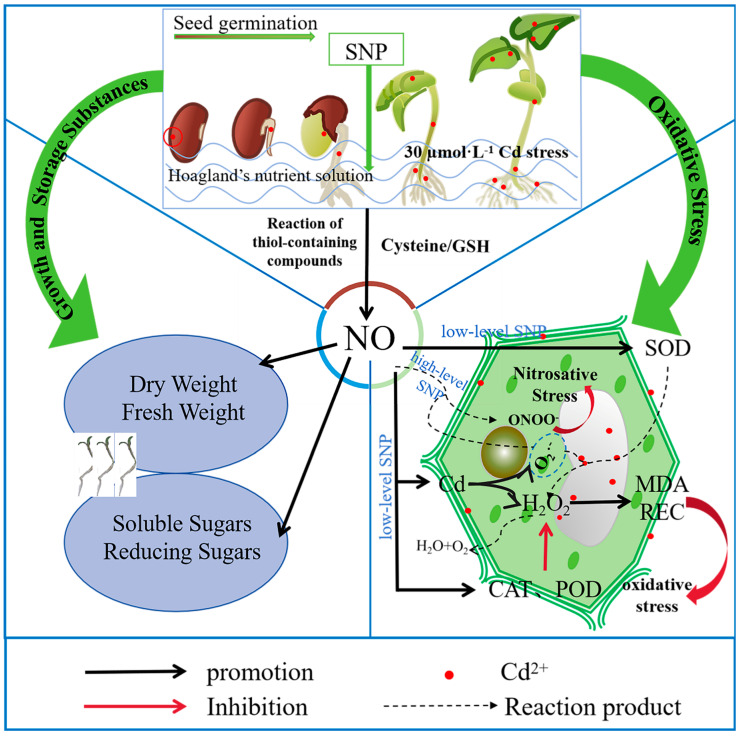
Proposed model for the role of NO in alleviating Cd-induced toxicity in alfalfa seedlings.

**Table 1 plants-14-03264-t001:** Effect of 200 µM SNP on growth and physiological metabolism of alfalfa. Seedlings were treated with nutrient solution (CK) and nutrient solution + 200 µM SNP (200SNP) for 5 days and cultured as described in Figure 1. Data are expressed as mean ± SD (*n* = 9). Values were compared using ANOVA, and when results were significant, the treatments were compared using Duncan’s test (*p* < 0.05). Values of a parameter labeled with different letters differ significantly.

Treatment	FW (g·100 Seedlings^−1^)	DW (g·100 Seedlings^−1^)	Germination Percentage (%)	Germination Index	Seedling Vigor Index (cm)	Germination Energy (%)
CK	1.40 ± 0.07 b	0.25 ± 0.01 b	55.68 ± 0.05 b	177.27 ± 8.55 b	353.34 ± 18.13 b	32.89 ± 1.25 b
200SNP	1.85 ± 0.04 a	0.36 ± 0.05 a	75.76 ± 2.22 a	294.58 ± 38.36 a	770.75 ± 112.11 a	54.91 ± 3.82 a
**Treatment**	**Root Length** **(mm)**	**Shoot Length** **(mm)**	**SS** **(µg·g^−1^ DW** **)**	**RS** **(µg·g^−1^ DW** **)**	**MDA** **(µmol** **·g^−1^ F** **W)**	**REC** **%**
CK	28.22 ± 2.12 b	18.00 ± 1.73 b	497.75 ± 10.33 a	558.82 ± 55.68 b	3.13 ± 0.09 a	87.23 ± 0.12 a
200 SNP	35.84 ± 0.92 a	24.52 ± 0.31 a	504.14 ± 16.23 a	699.18 ± 33.97 a	1.36 ± 0.12 b	85.64 ± 5.52 a
**Treatment**	**H_2_O_2_** **(µmol** **·g^−1^ F** **W)**	**O_2_·^−^** **(µmol** **·g^−1^ F** **W)**	**POD** **(U·g^−1^ FW·min^−1^** **)**	**SOD** **(U·g^−1^ FW·min^−1^** **)**	**CAT** **(U·g^−1^ FW·min^−1^** **)**	**APX** **(U·g^−1^ FW·min^−1^** **)**
CK	39.37 ± 2.49 a	0.14 ± 0.01 a	240.00 ± 0.80 a	70.53 ± 2.06 a	1.99 ± 0.02 a	23.33 ± 2.89 a
200SNP	12.41 ± 1.78 b	0.07 ± 0.01 b	176.69 ± 8.95 b	59.89 ± 2.29 b	1.44 ± 0.01 b	17.95 ± 1.02 b

## Data Availability

The original contributions presented in the study are included in the article/Appendix A. Further inquiries can be directed to the corresponding author.

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
