# Peer review of "Exogenous Nitric Oxide Promotes the Growth and Cadmium Accumulation of Alfalfa (Medicago sativa) Seedlings Under Cadmium Stress"

_plants, 2025, doi:10.3390/plants14213264_

Round 1
Reviewer 1 Report
Comments and Suggestions for Authors
The manuscript is devoted to the study of the effect of sodium nitroprusside, as an NO donor, on the manifestation of the toxic effects of Cd on Medicago sativa. The following comments have arisen when analyzing the manuscript:
- The plants were grown in Petri dishes on filter paper. This design of the experiment is not optimal, since filter paper can sorb Cd, as a result of which the real active concentration will be significantly lower than the declared one, and it is very difficult to normalize Cd concentration in such case. In addition, since a small amount of nutrient solution was added, the plant roots may not have been completely immersed in it (for example, see Fig. 1, variant 25SNP+Cd), which is not optimal for Cd absorption. Under such experimental design, the shoots of the seedlings also come into contact with the solution, which is not sufficiently correct from a physiological point of view. As a result, different seedlings of the same variant, as well as from different variants, will be in different conditions in terms of Cd and sodium nitroprusside uptake. Since this phenomenon can be observed in different numbers of seedlings to different degrees in different Petri dishes, it is difficult to correctly compare different treatments.
- A significant drawback of the experimental design is the lack of treatments with only sodium nitroprusside in the absence of Cd, which does not allow identifying the effect of NO on plants. These additional control treatments are mandatory for the correct conduct of the studies on the combined effects of metals and other substances, including NO [e.g. Kazemi et al., 2010; Kotapati et al., 2017; Rizwan et al., 2018; Soliman et al., 2019; Qin et al., 2022, Abbas et al., 2023]. Insufficiently correct experimental design is a significant drawback of this work.
- Throughout the text, it is necessary to check the consistency of the description with the data given in the figures. For example, the Authors write that “After 30 μM CdCl2 treatment, compared with the CK, the higher REC was from the first day to the fifth day, the higher production of H2O2 and O2•- were from the first day to the 4th day (P < 0.05)” (lines 118, 120), which for REC and H2O2 does not correspond to the data in Fig. 4 for day 2. Similarly, “After treatments with Cd and higher concentration of SNP (200 or 300 μM), H2O2 and O2•- levels gradually decreased to values similar to or below those in the CK” does not fully correspond to Fig. 4, for example the data for the plants after 4 days of incubation. “The activity of SOD reached the highest value when the SNP concentration was 200 μM” (lines 138-139) does not fully correspond to the data in Fig. 5.
- Part of the text in Fig. 9 is written in very small font and is difficult to read.
Author Response
|
Comments 1: The plants were grown in Petri dishes on filter paper. This design of the experiment is not optimal, since filter paper can sorb Cd, as a result of which the real active concentration will be significantly lower than the declared one, and it is very difficult to normalize Cd concentration in such case. In addition, since a small amount of nutrient solution was added, the plant roots may not have been completely immersed in it (for example, see Fig. 1, variant 25SNP+Cd), which is not optimal for Cd absorption. Under such experimental design, the shoots of the seedlings also come into contact with the solution, which is not sufficiently correct from a physiological point of view. As a result, different seedlings of the same variant, as well as from different variants, will be in different conditions in terms of Cd and sodium nitroprusside uptake. Since this phenomenon can be observed in different numbers of seedlings to different degrees in different Petri dishes, it is difficult to correctly compare different treatments. |
|
Response 1: Thank you for pointing this out. Seedlings were cultivated as described by He et al. (2014). This method was also applied by Mohammad et al. (2024). In addition, to avoid any change in the concentration, the test solutions were renewed every day. He, J.; Ren, Y.; Chen, X.; Chen, H. Protective roles of nitric oxide on seed germination and seedling growth of rice (Oryza sativa L.) under cadmium stress. Ecotoxicol. Environ. Saf. . 2014, 108, 114-119. Mohammad S. R., Farjana R., Shaila S. T., Naffz B.,Md T., Arif Hasan K. R., Md A. H., Yang X. H., Yoshiyuki M., Marian B. Proline and glycine betaine: A dynamic duo for enhancing salt stress resilience in maize by regulating growth, Stomatal size, and Oxidative stress responses. Plant stress, 2024, 14, 100563 |
|
Comments 2: A significant drawback of the experimental design is the lack of treatments with only sodium nitroprusside in the absence of Cd, which does not allow identifying the effect of NO on plants. These additional control treatments are mandatory for the correct conduct of the studies on the combined effects of metals and other substances, including NO [e.g. Kazemi et al., 2010; Kotapati et al., 2017; Rizwan et al., 2018; Soliman et al., 2019; Qin et al., 2022, Abbas et al., 2023]. Insufficiently correct experimental design is a significant drawback of this work. |
|
Response 2: Thank you for pointing this out.We have conducted experiments using the solely SNP, NO scavengers (cPTIO), NOS inhibitor (L-NAME) and NR inhibitor (NaN3) to validate the role of NO, and the results will be presented in another paper we are currently writing. The following figure shows some experimental data. As for this, we also have made a supplementary explanation in the form of future research prospects in the “4 Conclusion” section (lines from 370 to 379 in the revision).
|
|
Comments 3: Throughout the text, it is necessary to check the consistency of the description with the data given in the figures. For example, the Authors write that “After 30 μM CdCl2 treatment, compared with the CK, the higher REC was from the first day to the fifth day, the higher production of H2O2 and O2•- were from the first day to the 4th day (P < 0.05)” (lines 118, 120), which for REC and H2O2 does not correspond to the data in Fig. 4 for day 2. Similarly, “After treatments with Cd and higher concentration of SNP (200 or 300 μM), H2O2 and O2•- levels gradually decreased to values similar to or below those in the CK” does not fully correspond to Fig. 4, for example the data for the plants after 4 days of incubation. “The activity of SOD reached the highest value when the SNP concentration was 200 μM” (lines 138-139) does not fully correspond to the data in Fig. 5. |
|
Response 3: We agree with this comment. Therefore,the consistency of the description with the data given in the figures has been carefully checked and revised in the “2. Results” section (lines from 137 to 172) |
|
Comments 4: Part of the text in Fig. 9 is written in very small font and is difficult to read. |
|
Response 4: Figure 9(Figure 8 in the revised) has been modified in accordance with the reviewers' suggestions in the “3. Discussion” section (line 306). |

Reviewer 2 Report
Comments and Suggestions for Authors
Authors Chen et al. presented interesting work on alfalfa under Cd stress alleviation by nitric oxide (NO). The overall presentation is good, and the manuscript may be accepted after minor revisions. The following points should be addressed:
- Line 29: Delete “At present” and rewrite the sentence for better clarity.
- The novelty of the study needs to be clarified. The role of NO in heavy metal stress mitigation has already been widely studied. The manuscript mentions ROS metabolism as a novel aspect, but this is also well established. Please highlight the unique contribution of this work compared to previous studies.
- Figure 1: A scale bar should be included.
- Figure resolution should be increased,, especially for Figs. 7 and 8
- Too many abbreviations are introduced at their first appearance, which makes the text difficult to follow. Please reduce abbreviations and avoid starting sentences with abbreviations (e.g., Line 34 in the Results section: “REC…”).
- This section needs improvement. The authors should provide more interpretation of why specific parameters increased or decreased, supported by their hypotheses or assumptions. For guidance, the authors can refer to the following paper: http://dx.doi.org/10.1111/ppl.70344 http://dx.doi.org/10.1016/j.stress.2024.100563
- This section should be more concise, focusing only on the core findings and providing future recommendations.
Author Response
|
Comments 1: Line 29: Delete “At present” and rewrite the sentence for better clarity. |
|
Response 1: Thank you for pointing this out. We agree with this comment. This sentence has been revised according to the reviewers' suggestions in the “1. Introduction” section (lines from 34 to 37). |
|
Comments 2: The novelty of the study needs to be clarified. The role of NO in heavy metal stress mitigation has already been widely studied. The manuscript mentions ROS metabolism as a novel aspect, but this is also well established. Please highlight the unique contribution of this work compared to previous studies. |
|
Response 2: Thank you for pointing this out.The unique contribution of this work compared to previous studies has been supplemented in “1. Introduction” section (lines from 79 to 85). |
|
Comments 3: Figure 1: A scale bar should be included. |
|
Response 3: We agree with this comment. Therefore,A scale bar has been included in Figure 1(lines from 111 to 115). |
|
Comments 4: Figure resolution should be increased, especially for Figs. 7 and 8 |
|
Response 4: Figure resolution has been increased for Figs. 7 and 8(lines from 211 to 220). |
|
Comments 5: Too many abbreviations are introduced at their first appearance, which makes the text difficult to follow. Please reduce abbreviations and avoid starting sentences with abbreviations (e.g., Line 34 in the Results section: “REC…”). |
|
Response 5: Thank you for pointing this out. The issue regarding abbreviations has been revised in accordance with the experts' suggestions (lines from 48 to 49). |
|
Comments 6: This section needs improvement. The authors should provide more interpretation of why specific parameters increased or decreased, supported by their hypotheses or assumptions. For guidance, the authors can refer to the following paper: http://dx.doi.org/10.1111/ppl.70344 http://dx.doi.org/10.1016/j.stress.2024.100563 |
|
Response 6: The “2. Results” section has been improved in accordance with the experts' suggestions (lines from 102 to 103, 105, 121 to 122, 128 to 130, 137 to 142). |
|
Comments 7: This section should be more concise, focusing only on the core findings and providing future recommendations.: |
|
Response 7: The “5. Conclusions” section has been improved in accordance with the experts' suggestions (lines from 362 to 379). |

Reviewer 3 Report
Comments and Suggestions for Authors
The manuscript entitled "Exogenous nitric oxide promotes the growth and cadmium accumulation of alfalfa (Medicago sativa L.) seedlings under cadmium stress" explores the effects of exogenous nitric oxide (NO), applied through sodium nitroprusside (SNP), on the early growth, reactive oxygen species (ROS) metabolism, and cadmium (Cd) accumulation of alfalfa seedlings exposed to Cd stress. The study investigates physiological and biochemical changes and identifies an optimal SNP concentration for alleviating Cd toxicity.
The manuscript is well-structured with comprehensive experimental work and useful data supporting the role of NO in modulating Cd stress tolerance in alfalfa seedlings. The use of multiple biochemical assays and statistical analyses adds robustness to the conclusions. However, the presentation requires improvements in clarity, conciseness, and formatting to enhance readability and scientific impact.
Major Comments:
- The similarity index is 41% which should be reduced further; please check the author’s guidelines for further details.
- The title is clear and descriptive, reflecting the study’s focus accurately.
- The abstract is comprehensive but could be made more concise by avoiding overly detailed experimental methods. Key findings should be mentioned with % difference as compare to control and at the end conclude the abstract by mentioning future strategies and prospective.
- The introduction gives good background, but it should be enhance by adding the knowledge gap, what is known and what still need to be highlighted (Research gap), additionally would benefit from a clearer statement of the study’s hypotheses and aims at the end.
- There are several typographical errors and formatting inconsistencies throughout the text that distract from the scientific content.
- The figures and tables are informative but some figure legends lack sufficient detail to allow independent interpretation. Please explain the abbreviation in footnote; how many replications, and what different lettering is meaning for; please explain properly;
Merge figure 7 and 8; and improve the pixel quality. - The dosage range for SNP treatment is extensive; justification for the selection of this range should be briefly discussed.
- Some sentences are overly long or convoluted and could be simplified to enhance clarity.
- The discussion sometimes repeats results instead of interpreting them in the context of the literature. Please improve the discussion section by adding appropriate mechanisms.
- The dual role of NO as both beneficial and harmful is well mentioned but could be more critically analyzed.
- The paper would benefit from clearly differentiating novel findings from confirmations of previously published studies.
- Methodological details should be concise but complete; some sections could be shortened, and redundant descriptions removed.
- Statistical methods are appropriate but reporting of exact p-values rather than just inequalities (P < 0.05) in some cases would improve rigor.
- The conclusion is well stated but could briefly mention future research directions.
- Authors have already added the schematic diagram, but still this schematic figure need to improve as visually present the proposed NO mechanisms enhancing Cd tolerance are not mentioned properly and improve the figure caption.
Line-to-Line Comments:
- “drastically” could be replaced with a more scientific descriptor like “significantly”.
- Explain abbreviations such as SS and RS where first introduced.
- Consider splitting the long sentence into two for readability.
- Define “REC” before use.
- The phrasing “was reported to regulate” can be stronger, e.g., “regulates”.
- Clarify how NO decreases root-to-shoot Cd translocation.
- “Other studies have shown that NO exacerbates” – elaborate with short explanation or examples.
- Provide clearer transition when discussing phytoremediation.
- State hypothesis clearly at the beginning of the study aims.
- Figure 1 caption requires more experimental detail on conditions.
- Separate explanation of MDA induction and its timing better.
- Sentence structure around “did not differ significantly from those” is awkward.
- PCA and correlation analysis require more interpretative context.
- “Similar results have been reported in Elymus dahuricus” - add reference number.
- Explanation of ONOO- effects could be shortened or linked to figure 9 earlier.
- Sentence about NO’s need to be at low levels is unclear; rephrase.
- Discuss SNP’s dual effects on Cd uptake with more mechanistic insight.
- Last sentence about future genetic studies needs to be in future directions section.
- Throughout Methods, some chemical names and abbreviations lack units or standard nomenclature.
Moderate English editing is required.
Author Response
|
Comments 1: The similarity index is 41% which should be reduced further; please check the author’s guidelines for further details. |
|
Response 1: Thank you for pointing this out. After self-checking, the similarity index has been reduced from 41% to 13%. |
|
Comments 2: The title is clear and descriptive, reflecting the study’s focus accurately. |
|
Response 2: Thanks very much, for your affirmation of the topic. |
|
Comments 3: The abstract is comprehensive but could be made more concise by avoiding overly detailed experimental methods. Key findings should be mentioned with % difference as compare to control and at the end conclude the abstract by mentioning future strategies and prospective. |
|
Response 3: We agree with this comment. Therefore,The “Abstract” section has been improved in accordance with the experts' suggestions (lines from 10 to 30). |
|
Comments 4: The introduction gives good background, but it should be enhance by adding the knowledge gap, what is known and what still need to be highlighted (Research gap), additionally would benefit from a clearer statement of the study’s hypotheses and aims at the end. |
|
Response 4: Thank you for pointing this out.Research gap has been supplemented in accordance with the experts' suggestions (lines from 82 to 96). |
|
Comments 5: There are several typographical errors and formatting inconsistencies throughout the text that distract from the scientific content. |
|
Response 5: Thank you for pointing this out.The typographical errors and formatting inconsistencies have been revised in accordance with the experts' suggestions (lines from 39 to40). |
|
Comments 6: The figures and tables are informative but some figure legends lack sufficient detail to allow independent interpretation. Please explain the abbreviation in footnote; how many replications, and what different lettering is meaning for; please explain properly; Merge figure 7 and 8; and improve the pixel quality. |
|
Response 6: A scale bar has been added to Figure 1, and the meanings of abbreviations, number of replications, and implications of different letterings have been explained in the footnote of each subfigure to enhance readability(lines from 111 to 115 ); Figures 7 and 8 have been merged as required, and the resolution of all images has been improved to 600 dpi.(lines from 211 to 220 ). |
|
Comments 7: The dosage range for SNP treatment is extensive; justification for the selection of this range should be briefly discussed. |
|
Response 7: Justification for the selection of SNP dosage range has been supplemented (lines from 311 to 313). |
|
Comments 8:Some sentences are overly long or convoluted and could be simplified to enhance clarity. |
|
Response 8:These long sentences have been simplified (lines 242 and 246, 286 to 290 in the revision). |
|
Comments 9: The discussion sometimes repeats results instead of interpreting them in the context of the literature. Please improve the discussion section by adding appropriate mechanisms. |
|
Response 9: The “3. Discussion” section has been improved in accordance with the experts' suggestions (lines from 230 to 240, 270 to 277). |
|
Comments 10: The dual role of NO as both beneficial and harmful is well mentioned but could be more critically analyzed. |
|
Response 10: It has been supplemented in the last paragraph of“3. Discussion”section (lines from 298 to 304 in the revision) |
|
Comments 11: The paper would benefit from clearly differentiating novel findings from confirmations of previously published studies. |
|
Response 11: The differentiation has been supplemented in the“3. Discussion”section (lines from 258 to 267 in the revision) |
|
Comments 12: Methodological details should be concise but complete; some sections could be shortened, and redundant descriptions removed. |
|
Response 12: The“Materials and Methods”section has been optimized and redundant descriptions have been removed.(lines from 330 to 352 in the revision) |
|
Comments 13: Statistical methods are appropriate but reporting of exact p-values rather than just inequalities (P < 0.05) in some cases would improve rigor. |
|
Response 13: The exact p-values have been supplemented in the“2. Results”section (lines from 200 to 210 in the revision) |
|
Comments 14: The conclusion is well stated but could briefly mention future research directions. |
|
Response 14: The “5. Conclusions” section has been improved in accordance with the experts' suggestions (lines from 363 to 379). |
|
Comments 15: Authors have already added the schematic diagram, but still this schematic figure need to improve as visually present the proposed NO mechanisms enhancing Cd tolerance are not mentioned properly and improve the figure caption. |
|
Response 15: The schematic diagram has been improved (Figure 8, lines 305). |
|
Comments 16:“drastically” could be replaced with a more scientific descriptor like “significantly”. |
|
Response 16: "drastically" in the 12th line of the abstract has been replaced with a more scientific descriptor, "significantly". |
|
Comments 17: Explain abbreviations such as SS and RS where first introduced. |
|
Response 17: The full names corresponding to these abbreviations have been added in the 16th line of the abstract and at the position where they first appear in the text (Line 122 of the Results section). |
|
Comments 18: Consider splitting the long sentence into two for readability. |
|
Response 18: These long sentences have been simplified (lines 242 and 246, 286 to 290 in the revision). |
|
Comments 19: Define “REC” before use. |
|
Response 19: A definition of "REC" has been added at the 49th line of the Introduction, where "REC" first appears. |
|
Comments 20: The phrasing “was reported to regulate” can be stronger, e.g., “regulates”. |
|
Response 20: The revision has been made in the 56th line of the Introduction. |
|
Comments 21: Clarify how NO decreases root-to-shoot Cd translocation. |
|
Response 21: It has been supplemented in the“2. Results”section (lines from 59 to 63, 270 to 277 in the revision) |
|
Comments 22: “Other studies have shown that NO exacerbates” – elaborate with short explanation or examples. |
|
Response 22: Example has been supplemented in the“1. Introduction”section (lines from 64 to 66 in the revision). One new reference has been added in“References”(lines from 426 to 428 in the revision). |
|
Comments 23: Provide clearer transition when discussing phytoremediation. |
|
Response 23: Transition has been supplemented in the“1. Introduction”section (lines from 70 to 73 in the revision). |
|
Comments 24: State hypothesis clearly at the beginning of the study aims. |
|
Response 24: Hypothesis has been supplemented in the“1. Introduction”section (lines from 87 to 89 in the revision). |
|
Comments 25: Figure 1 caption requires more experimental detail on conditions. |
|
Response 25: Experimental detail on conditions has been supplemented in Figure 1 (lines 111 to 115 in the revision). |
|
Comments 26: Separate explanation of MDA induction and its timing better. |
|
Response 26: Explanation of MDA has been supplemented in the“2. Results” section (lines from 137 to 142 in the revision). |
|
Comments 27: Sentence structure around “did not differ significantly from those” is awkward. |
|
Response 27: The sentence structure has been revised in Lines 145-146 of the Results section. |
|
Comments 28: PCA and correlation analysis require more interpretative context. |
|
Response 28: PCA and correlation analysis has been supplemented in the“2. Results” and “3. Discussion” sections (lines from 186 to 187, 286 to 292 in the revision). |
|
Comments 29: Similar results have been reported in Elymus dahuricus” - add reference number. |
|
Response 29: After discussion, it was deemed that this sentence is not suitable to be placed here and has therefore been deleted (lines from 223 to 224 in the revision). |
|
Comments 30: Explanation of ONOO- effects could be shortened or linked to figure 9 earlier. |
|
Response 30: Explanation of ONOO- effects has been shortened in the“3. Discussion” sections (lines from 265 to 266 in the revision). |
|
Comments 31: Sentence about NO’s need to be at low levels is unclear; rephrase. |
|
Response 31: Thank you for your suggestions. We agree with your views and have made the revisions (lines from 252 to 253 in the revision). |
|
Comments 32: Discuss SNP’s dual effects on Cd uptake with more mechanistic insight. |
|
Response 32: It has been supplemented in the “3. Discussion” sections (lines from 260 to 265, 269 to 282 in the revision). |
|
Comments 33: Last sentence about future genetic studies needs to be in future directions section. |
|
Response 33: It has been moved to future directions section (the last paragraph of conclusion) (lines from 377 to 379). |
|
Comments 34: Throughout Methods, some chemical names and abbreviations lack units or standard nomenclature. |
|
Response 34: Thank you for your suggestions. We agree with your views and have made the revisions (lines from 308 to 361). |

Reviewer 4 Report
Comments and Suggestions for Authors
The effects of RNS in alfalfa seedling under Cd stress were already examined e.g.:
Li, L., Wang, Y. & Shen, W. Roles of hydrogen sulfide and nitric oxide in the alleviation of cadmium-induced oxidative damage in alfalfa seedling roots. Biometals 25, 617–631 (2012). https://doi-1org-1x3z6aval077c.han.amu.edu.pl/10.1007/s10534-012-9551-9
Fang L, Ju W, Yang C, Duan C, Cui Y, Han F, Shen G, Zhang C. Application of signaling molecules in reducing metal accumulation in alfalfa and alleviating metal-induced phytotoxicity in Pb/Cd-contaminated soil. Ecotoxicol Environ Saf. 2019 Oct 30;182:109459. doi: 10.1016/j.ecoenv.2019.109459. Epub 2019 Jul 22. PMID: 3134459.
However, the manuscript adds to new data to the topic. There are some issues, which should be modified/clarified before the publication of the results.
- The text coverage with other publications is quite high. I would suggest to re-write the introduction in own words to avoid overlap with text from other publications.
- The study is missing the solely SNP treatment. The effect of SNP in different concentrations should be included at least in relation to the growth parameters, to show that the donor itself has no effect on the seedlings. This could be included as supplementary data.
- It would be recommendable to include also variant with scavenger of NO to confirm its effects, e.g. in chosen variants and for chosen analysis showing highest effects of SNP.
- In the abstract, it would be better to write “The donor of NO, SNP, mitigated membrane peroxidation damage caused by Cd stress in alfalfa by enhancing activities of catalase (CAT), ascorbate peroxidase (APX), superoxide dismutase (SOD) and peroxidase (POD), so as to eliminate ROS and improve Cd resistance. In addition, treatment with SNP promoted Cd accumulation” . Otherwise the sentence suggests that stimulation of the antioxidant enzymes leads to higher Cd accumulation.
- In the results section, the Fig.1 is not very clear. I would suggest to include another picture with seedling from all variants on one photo, on higher magnification and made on a different angle.
- I would suggest to change the term “exogenous NO” to “NO donor”
- Figure 9 is not very informative. The abbreviation SS/RS could be written in full. It is not clear what is the reason for including the scheme of membrane in this particular place. It is also not clear what is the meaning of lines marked in blue, green and brown. I would suggest to leave just the scheme of the experiments with an arrow pointing from SNP to NO and three arrows pointing to the influence of NO on the oxidative/antioxidant response, Cd uptake and sugar metabolism.
- In conclusions: This section would need language correction. I would also suggest to move the second paragraph to the discussion section, or point it out as future research plans (if such plans are under consideration).
Minor comments:
- It would be good to mention in the introduction that ROS can also exert beneficial functions.
- In line 57: the word “expression” is doubled
- Please include Latin and common names of all mentioned plant species.
- Some terms could be replaced e.g. “appearance of the seedlings” to “phenotype of the seedlings” (line 90), “SNP produces” to “SNP confers” (lines 96-97)
- Please include also the information that SD is present on the graphs in the description of figures.
- Some sentences sentence are very long and could be divided it into two or three sentences (e.g. lines 214-219; 248-254).
- In conclusions: I would avoid starting the sentence with “30 µM”, maybe better start “Present research shows that 30 µM Cd….”
Author Response
|
Comments 1: The text coverage with other publications is quite high. I would suggest to re-write the introduction in own words to avoid overlap with text from other publications. |
|
Response 1: Thank you for pointing this out. After self-checking, the similarity index has been reduced from 41% to 13%. |
|
Comments 2: The study is missing the solely SNP treatment. The effect of SNP in different concentrations should be included at least in relation to the growth parameters, to show that the donor itself has no effect on the seedlings. This could be included as supplementary data. It would be recommendable to include also variant with scavenger of NO to confirm its effects, e.g. in chosen variants and for chosen analysis showing highest effects of SNP. |
|
Response 2: Thanks for the reviewers' comments. We have conducted experiments using the solely SNP, NO scavengers (cPTIO), NOS inhibitor (L-NAME) and NR inhibitor (NaN3) to validate the role of NO, and the results will be presented in another paper we are currently writing. The following figure shows some experimental data. As for this, we also have made a supplementary explanation in the form of future research prospects in the “4 Conclusion” section (lines from 370 to 379 in the revision).
|
|
Comments 3: In the abstract, it would be better to write “The donor of NO, SNP, mitigated membrane peroxidation damage caused by Cd stress in alfalfa by enhancing activities of catalase (CAT), ascorbate peroxidase (APX), superoxide dismutase (SOD) and peroxidase (POD), so as to eliminate ROS and improve Cd resistance. In addition, treatment with SNP promoted Cd accumulation” . Otherwise the sentence suggests that stimulation of the antioxidant enzymes leads to higher Cd accumulation. |
|
Response 3: We agree with this comment. This sentence has been revised in “Abstract” section (lines from 17 to 20 in the revision). |
|
Comments 4: In the results section, the Fig.1 is not very clear. I would suggest to include another picture with seedling from all variants on one photo, on higher magnification and made on a different angle. |
|
Response 4: Thank you for pointing this out. The clarity of Fig. 1 has been enhanced to 600 dpi, a scale bar has been added to each subfigure, and the scaling of all scale bars has been standardized to facilitate comparative observation. (lines from 111 to 115 in the revision). |
|
Comments 5: I would suggest to change the term “exogenous NO” to “NO donor” |
|
Response 5: Thank you for your suggestion. We have replaced "exogenous NO" appearing in the manuscript with "NO donor" or "SNP" at Line 14, Line 112, Line 117, and 11 other locations (totaling 14 places). |
|
Comments 6: Figure 9 is not very informative. The abbreviation SS/RS could be written in full. It is not clear what is the reason for including the scheme of membrane in this particular place. It is also not clear what is the meaning of lines marked in blue, green and brown. I would suggest to leave just the scheme of the experiments with an arrow pointing from SNP to NO and three arrows pointing to the influence of NO on the oxidative/antioxidant response, Cd uptake and sugar metabolism. |
|
Response 6: Extensive revisions have been made to Figure 9 as required.(lines 305 from to 306). |
|
Comments 7: In conclusions: This section would need language correction. I would also suggest to move the second paragraph to the discussion section, or point it out as future research plans (if such plans are under consideration). |
|
Response 7: The “Conclusions” section has been revised (lines from 363 to 379 in the revision) |
|
Comments 8: It would be good to mention in the introduction that ROS can also exert beneficial functions. |
|
Response 8:The beneficial functions of ROS has been supplemented in the“3. Discussion”sections (lines from 49 to 52 in the revision).One new reference has been added in“References”(lines from 365 to 366 in the revision). |
|
Comments 9: In line 57: the word “expression” is doubled |
|
Response 9: The second "expression" in this line has been deleted.(lines from 62 to 64 in the revision) |
|
Comments 10: Please include Latin and common names of all mentioned plant species. |
|
Response 10: The Latin names and common names of all mentioned plant species have been supplemented as required.(lines from 71 to 73, 226 to 227, 247 to 249, 269 to 270 in the revision) |
|
Comments 11: Some terms could be replaced e.g. “appearance of the seedlings” to “phenotype of the seedlings” (line 90), “SNP produces” to “SNP confers” (lines 96-97) |
|
Response 11: Revisions have been made in accordance with the suggestions.(lines 103 and 109 in the revision) |
|
Comments 12: Please include also the information that SD is present on the graphs in the description of figures. |
|
Response 12: As requested, the information indicating that SD (Standard Deviation) is present on the graphs has been added to the descriptions of Figure 2 to Figure 6.(lines 116 to 120, 134, 156 to 157, 171 to 172, 183 to 184 in the revision) |
|
Comments 13: Some sentences are very long and could be divided it into two or three sentences (e.g. lines 214-219; 248-254). |
|
Response 13: Revisions have been made in line with the suggestion(lines 242 and 246, 286 to 290 in the revision). |
|
Comments 14: In conclusions: I would avoid starting the sentence with “30 µM”, maybe better start “Present research shows that 30 µM Cd…. |
|
Response 14: This sentence has been revised (lines from 362 to 364 in the revision) |

Round 2
Reviewer 1 Report
Comments and Suggestions for Authors
Insufficiently correct experimental design is a significant drawback of this work.
I am still convinced that the design of the experiment is not appropriate, since filter paper can sorb Cd (to a different degree depending on its type), as a result of which the real active concentration of Cd will be significantly lower than the declared one. Moreover, it is very difficult to normalize Cd concentration in such case. In addition, since the roots were not completely immersed in nutrient solution (for example, see Fig. 1), Cd absorption might have differed significantly between the plants within and among variants. Moreover, the shoots of the seedlings also came into contact with the solution, which is not sufficiently correct from a physiological point of view. If different seedlings of the same variant, as well as from different variants, were in different conditions in terms of Cd and sodium nitroprusside uptake, it is impossible to correctly compare different treatments.
The additional control treatments (e.g. with only sodium nitroprusside in the absence of Cd ) are mandatory for the correct conduct of the studies on the combined effects of metals and other substances, including NO. It is important to provide these data in the current manuscript.
Author Response
|
Comments 1: I am still convinced that the design of the experiment is not appropriate, since filter paper can sorb Cd (to a different degree depending on its type), as a result of which the real active concentration of Cd will be significantly lower than the declared one. Moreover, it is very difficult to normalize Cd concentration in such case. |
|
Response 1: We highly agree with your suggestion. The filter paper does indeed cause the concentration of Cd to be lower than the concentration set in our experiment. In order to reduce the errors caused by filter paper, in our experiment, first, all treatments used the same specification of filter paper (such as the number of layers, material) to ensure that the adsorption capacity of Cd by the filter paper in all treatments was consistent , so the Cd concentration levels in all treatments remained consistent; second, Cd stress was added in the form of CdClâ‚‚ solution, which could avoid concentration deviation caused by water absorption of filter paper; third, we used three replicates of each treatment every day, and took the average value to reduce errors; fourth, to avoid any change in the Cd concentration, the test solutions were renewed every day. We have detailed explanations of these in "4. Materials and Methods" section (lines from 346 to 349, 353 to 354, 357 to 358 in the revision). Therefore, the condition equilibrium of all treatments in our experiment can avoid systematic error. Even if the concentration of Cd is lower than the set concentration, the trend of the data will not change. Thanks very much for your reminder, in future experiments, we must refer to this suggestion and take some measures to effectively reduce the influence of filter paper on Cd concentration, such as using glass fiber filter paper instead of ordinary filter paper. Thank you very much again. The deficiencies in the use of filter paper and the improvement methods have been supplemented in the form of future research prospects in the “5 Conclusion” section (lines from 418 to 423 in the revision).
|
|
Comments 2: In addition, since the roots were not completely immersed in nutrient solution (for example, see Fig. 1), Cd absorption might have differed significantly between the plants within and among variants. Moreover, the shoots of the seedlings also came into contact with the solution, which is not sufficiently correct from a physiological point of view. If different seedlings of the same variant, as well as from different variants, were in different conditions in terms of Cd and sodium nitroprusside uptake, it is impossible to correctly compare different treatments. |
|
Response 2: We sincerely apologize for any misunderstanding caused by our unclear explanation. Figure 1 in the manuscript shows the seedlings growth on the fifth day after different treatments. In order to see the differences among various treatments more clearly, only one seedling was placed in every petri dish in Figure 1. It is indeed that we did not express clearly. We have provided a clear description of Figure 1 in "2.1. Seedling Growth" section (lines from 106 to 108, 117 to 118 in the revision). Images of growth conditions of all the seedlings has been supplemented in the form of supplementary materials (Figure S1: The growth conditions of all the seedlings treated with different concentration of SNP under 30 µM Cd stress.) (lines from 425 to 427 in the revision) We highly agree with your suggestion. There are indeed differences in the absorption of Cd among different individuals. Some seedlings absorb more through their root systems, while others also absorb Cd through their buds. In order to reduce the errors caused by individual differences, in our experiments, when measuring various indicators, we used mixed samples. For instance, when measuring FW, DW and the content of Cd, we randomly selected 100 seedlings for testing, moreover, each treatment has three repetitions. This sampling method can avoid errors caused by individual differences. We have detailed explanations of these in "4. Materials and Methods" section (lines from 371 to 372, 389 to 393 in the revision).
|
|
Comments 3: The additional control treatments (e.g. with only sodium nitroprusside in the absence of Cd) are mandatory for the correct conduct of the studies on the combined effects of metals and other substances, including NO. It is important to provide these data in the current manuscript. |
|
Response 3: We highly agree with your suggestion. According to your suggestion, the additional control treatments with only SNP (200 µM, the optimal concentration) in the absence of Cd has been supplemented. The effect of 200 µM SNP alone on growth and physiological metabolism of alfalfa has been showed in the form of table (Table 1), at the same time, a result analysis of this part of the data was also provided in "2. Results " and "3. Discussion" section (lines from 195 to 214, 305 to 308 in the revision). Experimental design, plants growth conditions and growth parameters determination methods have been supplemented in "4. Materials and Methods" section (lines from 360 to 369 in the revision). A reference also has been supplemented in "References" section (line 534 in the revision). |

Reviewer 2 Report
Comments and Suggestions for Authors
Thank you, authors, for revising the manuscript accordingly. It is now much improved and can be accepted.
Author Response
Comment 1: Thank you, authors, for revising the manuscript accordingly. It is now much improved and can be accepted.
Response : Thank you very much for your comments and for your work to improve our paper quality.

Reviewer 3 Report
Comments and Suggestions for Authors
The authors have significantly improved the quality of the manuscript in response to the comments and suggestions provided by the reviewer; the manuscript is now suitable for publication. However, before final publications, authors are requested to add the hypothesis of the study at the end of the introduction section.
Author Response
|
Comments 1: The authors have significantly improved the quality of the manuscript in response to the comments and suggestions provided by the reviewer; the manuscript is now suitable for publication. However, before final publications, authors are requested to add the hypothesis of the study at the end of the introduction section. |
|
Response 1: Thank you very much for your comments and for your work to improve our paper quality. According to your suggestions, the hypothesis of the study has been added at the end of "1. Introduction" section (lines from 86 to 91 in the revision). |

Reviewer 4 Report
Comments and Suggestions for Authors
Thank you for addressing most of the comment. However, in my opinion the manuscript still needs substantial modifications before its publication.
- In my opinion the lines 24-30 in the abstract should be moved to the end part of the discussion section as future plants of the research.
- Fig. 1 is still not very representative and informative. By this angle the seedlings growth and morphology can not be compared. If the authors are unable to make a new picture I would suggest to skip it and also skip the first paragraph of the results.
- The descriptions of the x and y axis of the figures (Fig. 2-6) as well as the names of the variants are not very clear. I would suggest to increase the font and the resolution.
- The sections 2.2 and 2.3 could be rechecked and rewritten as in some cases it is difficult to follow the text. Some sentences are too long or unclear.
- The discussion could be rechecked. Some terms are not entirely scientifically proper e.g. “NO defend cell membrane”, “antioxidase”, “cell suffering”
- Fig. 8 is still not very clear e.g. the arrow from alfalafa seeds pointing to growth parameters suggest that these were assessed in untreated seeds. It would be better to point both arrows from the drawing of the seedlings. The term nitrosative stress is included in the figure but it was actually not measured. Why are the arrows from SOD and REC pointing to SOD, CAT and POD?
- In the material and methods sections 4.3 and 4.4 should be elaborated. How was the amount of Cd assessed?
- In conclusions, the second paragraph should be moved to the discussion section as it is not a conclusion based on the results. I would omit the part on filter paper as it is rather clear that depending on the medium the uptake of Cd would differ. Similarly, if the plants would be grown in agar, soil or hydroponically the level of Cd absorption by plant would differ.
Author Response
Comment 1: Thank you for addressing most of the comment. However, in my opinion the manuscript still needs substantial modifications before its publication.
In my opinion the lines 24-30 in the abstract should be moved to the end part of the discussion section as future plants of the research.
Response: Thank you very much for your comments and for your work to improve our paper quality. This part has been moved to the end part of the discussion section (lines from 324 to 333 in the revision).
Comment 2: Fig. 1 is still not very representative and informative. By this angle the seedlings growth and morphology can not be compared. If the authors are unable to make a new picture I would suggest to skip it and also skip the first paragraph of the results.
Response: Thank you very much for your suggestion. According to your suggestions, we have made a new picture to replace old one in Fig. 1 (line 110 in the revision).
Comment 3: The descriptions of the x and y axis of the figures (Fig. 2-6) as well as the names of the variants are not very clear. I would suggest to increase the font and the resolution.
Response: According to your suggestions, the font and the resolution of the figures (Fig. 2-6) have been increased (lines 125, 130, 147, 162, 177 in the revision).
Comment 4: The sections 2.2 and 2.3 could be rechecked and rewritten as in some cases it is difficult to follow the text. Some sentences are too long or unclear.
Response: According to your suggestions, the sections 2.2 and 2.3 have been rechecked and rewritten carefully (lines from 137 to 146, 152 to158, 169 to176 in the revision).
Comment 5: The discussion could be rechecked. Some terms are not entirely scientifically proper e.g. “NO defend cell membrane”, “antioxidase”, “cell suffering”
Response: Thanks for your good suggestion. According to your suggestions, “NO defend cell membrane” has been replaced by “NO protect cell membranes from lipid peroxidation damage” (lines 264 in the revision), all “antioxidase” have been replaced by “antioxidant enzyme” (lines 35, 84, 90, 159, 195, 247, 268, 447 in the revision), “cell suffering” have been replaced by “cell damage” (lines 271, 314 in the revision).
Comment 6: Fig. 8 is still not very clear e.g. the arrow from alfalafa seeds pointing to growth parameters suggest that these were assessed in untreated seeds. It would be better to point both arrows from the drawing of the seedlings. The term nitrosative stress is included in the figure but it was actually not measured. Why are the arrows from SOD and REC pointing to SOD, CAT and POD?
Response: According to your suggestions, Fig. 8 has been revised (line 335 in the revision).
Comment 7: In the material and methods sections 4.3 and 4.4 should be elaborated. How was the amount of Cd assessed?
Response: According to your suggestions, “4.3 Lipid Peroxidation Determination”, “4.4 Antioxidant Enzyme Extraction and Assays” and “4.6 Cd concentration Determination” have been elaborated (lines from 376 to 397, 399 to 417, 425 to 433 in the revision).
Comment 8: in conclusions, the second paragraph should be moved to the discussion section as it is not a conclusion based on the results. I would omit the part on filter paper as it is rather clear that depending on the medium the uptake of Cd would differ. Similarly, if the plants would be grown in agar, soil or hydroponically the level of Cd absorption by plant would differ.
Response: Thank you very much for such a good suggestion. According to your suggestions, the second paragraph has been moved to the "3. Discussion" section (lines from 324 to 333 in the revision).
The part on filter paper has been omitted in conclusions.

Round 3
Reviewer 3 Report
Comments and Suggestions for Authors
The authors have significantly improved the quality of the manuscript in response to the comments and suggestions provided by reviewers; the manuscript is now suitable for publication.
Author Response
Comment 1: The authors have significantly improved the quality of the manuscript in response to the comments and suggestions provided by reviewers; the manuscript is now suitable for publication.
Response: Thank you very much for your comments and for your work to improve our paper quality.

Reviewer 4 Report
Comments and Suggestions for Authors
Thank you for all the modifications.
Some minor comments:
Line 72: should be “is” instead of “has”
Line 82-85: These are rather the actual obtained results than a hypothesis. Mayber better “Therefore, the hypothesis of this study is that an appropriate concentration of SNP will exert protective functions in alfalfa exposed to Cd”.
Line 80: maybe instead of “NO maybe has a dual function in plants responses to Cd” better “NO exhibited dual functions in plants responses to Cd”
Line 246: should be “which damage cell membranes” instead of “which hurt cell membranes”
Line 251: I would suggest to use word “indicate” instead of “show” (in “Our results showed that SNP can strongly stimulate amylases…”) as the actual activity of amylases was not studied.
Line 258: I would suggest to delete the term “As we known,”
Line 275: Maybe better: “In our study, alfalfa had a better ability of accumulating Cd when treated with SNP (Figure 6), in accord with 275 results of Wang et al.”
Line 279: Maybe better “However, this positive, in relation to phytoremediation, role of NO depends on the concentration of SNP.”
Line 282-283: Maybe better: “It may be dependent on the fact, NO can react with ROS to produce more toxic oxides such as ONOO-“
Line 294: Better “In our study, 200 μM SNP did not exhibit negative effects on alfalfa but instead promoted its growth in the absence of Cd by enhancing breakdown”
Lines 299-301: Better “These different results indicate that NO may have a dual action on heavy metal accumulation in plants by regulating expression of genes related to heavy metal absorption and transport.”
Line 329: “NO synthase (NOS)” should be changes to “possible enzyme with NO synthase activity” as to the best of my knowledge NO synthase has not been yet identified in plants
Line 433: Maybe instead of „product of Cd content and biomass is total Cd accumulation” better “The total Cd accumulation was calculated on the basis of Cd concentration and plant biomass”
Lines 424-433: Was any reference material used for the determination of Cd content?
Author Response
|
Comments 1: Line 72: should be “is” instead of “has” |
|
Response 1: Thank you for pointing out this issue! You're absolutely right and "has" in line 72 has be changed to "is" to ensure grammatical accuracy (line 72 in the revision). |
|
Comments 2: Line 82-85: These are rather the actual obtained results than a hypothesis. Mayber better “Therefore, the hypothesis of this study is that an appropriate concentration of SNP will exert protective functions in alfalfa exposed to Cd”. |
|
Response 2: Thank you very much for your suggestion. According to your suggestions, we have revised the hypothesis of this study (lines from 82 to 84 in the revision). |
|
Comments 3: Line 80: maybe instead of “NO maybe has a dual function in plants responses to Cd” better “NO exhibited dual functions in plants responses to Cd” |
|
Response 3: According to your suggestions, we have revised “NO maybe has a dual function in plants responses to Cd” line 80 to "NO exhibited dual functions in plants' responses to Cd." (line 80 in the revision). |
|
Comments 4: Line 246: should be “which damage cell membranes” instead of “which hurt cell membranes” |
|
Response 4: According to your suggestions, we have replaced "hurt" with "damage" (line 245 in the revision). |
|
Comments 5: Line 251: I would suggest to use word “indicate” instead of “show” (in “Our results showed that SNP can strongly stimulate amylases…”) as the actual activity of amylases was not studied. |
|
Response 5: Thank you for your insightful suggestion! We have replaced "show" with "indicate" to make the expression more rigorous and accurate (line 250 in the revision). |
|
Comments 6: Line 258: I would suggest to delete the term “As we known,” |
|
Response 6: According to your suggestions, “As we known,” has been deleted (line 257 in the revision). |
|
Comments 7: Line 275: Maybe better: “In our study, alfalfa had a better ability of accumulating Cd when treated with SNP (Figure 6), in accord with 275 results of Wang et al.” |
|
Response 7: According to your comments, we have revised the content of this sentence (lines from 274 to 275 in the revision). |
|
Comments 8: Line 279: Maybe better “However, this positive, in relation to phytoremediation, role of NO depends on the concentration of SNP.” |
|
Response 8: Thank you very much for such a good suggestion. We have revised the relevant expression to emphasize the association between the positive role of NO and phytoremediation (lines from 278 to 279 in the revision). |
|
Comments 9: Line 282-283: Maybe better: “It may be dependent on the fact, NO can react with ROS to produce more toxic oxides such as ONOO-“. |
|
Response 9: According to your comments, we have revised the content of this section (lines from 282 to 283 in the revision). |
|
Comments 10: Line 294: Better “In our study, 200 μM SNP did not exhibit negative effects on alfalfa but instead promoted its growth in the absence of Cd by enhancing breakdown” |
|
Response 10: Thank you for your insightful suggestion! According to your comments, we have revised the content of this section (lines from 294 to 295 in the revision). |
|
Comments 11: Lines 299-301: Better “These different results indicate that NO may have a dual action on heavy metal accumulation in plants by regulating expression of genes related to heavy metal absorption and transport.” |
|
Response 11: According to your comments, we have revised the content of this section (lines from 299 to 301 in the revision). |
|
Comments 12: Line 329: “NO synthase (NOS)” should be changes to “possible enzyme with NO synthase activity” as to the best of my knowledge NO synthase has not been yet identified in plants |
|
Response 12: According to your comments, we have revised the content of this section (line 326 in the revision). |
|
Comments 13: Line 433: Maybe instead of “product of Cd content and biomass is total Cd accumulation” better “The total Cd accumulation was calculated on the basis of Cd concentration and plant biomass” |
|
Response 13: Thank you for your insightful suggestion! According to your comments, we have revised the content of this section (lines from 435 to 436 in the revision). |
|
Comments 14: Lines 424-433: Was any reference material used for the determination of Cd content? |
|
Response 14: Thank you for your question regarding the Cd content determination! In the determination of Cd content in this experiment, reference materials (HTSB-2 and NST-2) were indeed used to ensure the accuracy and reliability of the Cd content determination results. We have supplemented this description in the revised manuscript (lines from 425 to 428 in the revision). |

Round 4
Reviewer 4 Report
Comments and Suggestions for Authors
Thank you for all the modifications.